# Heavy N$^+$ ion transfer in doubly charged N$_2$Ar van der Waals cluster

XiaoLong Zhu [1,2,6], XiaoQing Hu [3,6], ShunCheng Yan[1,2], YiGeng Peng[3], WenTian Feng[1], DaLong Guo[1,2], Yong Gao[1], ShaoFeng Zhang[1,2], Amine Cassimi[4], JiaWei Xu[1,2], DongMei Zhao[1], DaPu Dong[1,2], Bang Hai[1,2], Yong Wu [3,5✉], JianGuo Wang[3] & X. Ma [1,2✉]

Van der Waals clusters are weakly bound atomic/molecular systems and are an important medium for understanding micro-environmental chemical phenomena in bio-systems. The presence of neighboring atoms may open channels otherwise forbidden in isolated atoms/molecules. In hydrogen-bond clusters, proton transfer plays a crucial role, which involves mass and charge migration over large distances within the cluster and results in its fragmentation. Here we report an exotic transfer channel involving a heavy N$^+$ ion observed in a doubly charged cluster produced by 1 MeV Ne$^{8+}$ ions: $(N_2Ar)^{2+} \rightarrow N^+ + NAr^+$. The neighboring Ar atom decreases the $N_2^{2+}$ barrier height and width, resulting in significant shorter lifetimes of the metastable molecular ion state $N_2^{2+}(X^1\Sigma_g^+)$. Consequently, the breakup of the covalent $N^+ - N^+$ bond, the tunneling out of the $N^+$ ion from the $N_2^{2+}$ potential well, as well as the formation of an $N - Ar^+$ bound system take place almost simultaneously, resulting in a Coulomb explosion of $N^+$ and $NAr^+$ ion pairs.

[1] Institute of Modern Physics, Chinese Academy of Sciences, 730000 Lanzhou, China. [2] University of Chinese Academy of Sciences, 100049 Beijing, China. [3] Institute of Applied Physics and Computational Mathematics, 100088 Beijing, China. [4] CIMAP, CEA/CNRS/ENSICAEN/UNICAEN, BP5133, 14070 Caen, France. [5] HEDPS, Center of Applied Physics and Technology, Peking University, 100871 Beijing, China. [6] These authors contributed equally: XiaoLong Zhu, XiaoQing Hu. ✉email: wu_yong@iapcm.ac.cn; x.ma@impcas.ac.cn

Van der Waals (vdW) interactions are ubiquitous in nature and important for many physical and chemical phenomena. A good understanding of intermolecular interaction is essential since weakly bond systems widely exist in biology and play a key role as a reactive intermediate in many processes. Weak bonds may be easily broken but they are very important because they enable to determine and stabilize the shape of biological molecules, such as stabilizing the secondary structure (e.g. alpha-helix and beta-pleated sheet) of proteins[1]. Dimers are good prototypes to investigate this weakly bound interaction. Indeed, atomic/molecular dimers have a typical weak binding energy in the order of meV and have a large distance between its two components usually above 7 a.u., which exceeds the covalent bond in molecules (~2 a.u.). Although weakly bound rare gas dimers have been studied extensively during the last two decades[2–15], studies of dimers consisting of one or two diatomic molecules remain rare, some examples can be found in refs. [16–22]. Therefore, it is essential to understand the structure and fragmentation/dissociation dynamics of charged molecular dimers at the level of molecular motion[2].

A prominent example is the interatomic Coulombic decay (ICD) theoretically predicted in 1997 (ref. [3]), where electronic excitation energy is efficiently transferred over a large internuclear distance within a rare gas dimer to allow for fast nonradiative relaxation. In 2003, ICD was first observed experimentally in $Ne_2$ rare gas cluster following inner-valence electron ionization[5]. Later, numerous experiments, where photons, electrons, and heavy ions were used as projectiles, have been performed to confirm the ICD and new dissociation mechanisms have been found in different rare gas dimers or larger clusters[6–21,23–28], such as radiative charge transfer[8,15] or electron-transfer-mediated decay[10,27]. For this kind of processes, the energy and charge transfer are mediated by virtual photon/Coulomb interaction, or electron transfer. In addition to the mechanisms mentioned above, an intermolecular proton transfer was reported in the molecular dimers, such as acetylene dimer[29], water cluster[23,30], and ammonia clusters[31]. In this process, a proton is transferred from a parent molecule to its partner molecule in a dimer which is bound together by hydrogen bond, and then, leading to a dissociation breakup. Since the discovery of the double helix form of DNA and the hypothesis of DNA mutation induced by proton transfer[32] more than 50 years ago, it has been recognized that proton transfer is crucial in many chemical and biological processes[33]. For example, a proton transfer reaction may produce a rare tautomer, which is regarded as the source of spontaneous mutations, and might lead to some important biochemistry processes related to other relevant diseases such as cellular aging and cancer[32,34]. On the other hand, there are excited-state proton transfer and multiple proton transfer processes which also play vital roles in biological systems. Nowadays, at the molecular level the simulations employing quantum chemistry could help us to have a thorough and detailed understanding of the mechanism of chemical and biological processes relevant to DNA, heredity, mutation, cancer, and green fluorescence protein, etc.[35].

In this article, we report a dissociation breakup mechanism of $N_2^{2+}Ar$ cluster induced by heavy $N^+$ ion transfer: $N_2^{2+}Ar \rightarrow N^+ + NAr^+$. Here, the strong covalent bond N–N breaks up and a new N–Ar bond is formed, due to $N^+$ transfer from $N_2^{2+}$ center to Ar in doubly charged $N_2^{2+}Ar$. To the best of our knowledge, such a heavy ion transfer process in vdW cluster has never been reported before and the consequent formation of $NAr^+$ is a novel scenario. It is intriguing to build bridges between the heavy ion transfer processes and biological processes at molecular level in the micro-environment of biological system as well as, for example, in the understanding of micro-mechanism of cancer therapy by heavy ion irradiations.

## Results

**Identification of heavy ion transfer channel.** In the present work, doubly charged cluster $(N_2Ar)^{2+}$ ions are produced by using 1 MeV $Ne^{8+}$ ions impact on neutral vdW cluster $N_2Ar$ target, and the fragment ions are detected in coincidence with the scattered ions. The information of time-of-flight (TOF) and positions of each charged fragment arriving at the detector is used to reconstruct their three-dimensional initial momentum vectors (see Methods). Figure 1 shows the two-dimensional correlation map between the TOFs of the first and the second fragment ions produced in $Ne^{8+}$–$N_2Ar$ collisions with one or two electrons stabilized on the projectile. At the top of horizontal axis the ion species and charge states corresponding to the TOF1 are labeled. Similarly, the ion species and charge states corresponding to the TOF2 (vertical axis) are also indicated. Usually, the different reaction channels can be identified according to the charge conservation before and after the collision. As a general rule for TOF correlation map, the slash lines are characteristics of two-body dissociation processes, while the extended area around the slash lines originates from the three-body or multi-body fragmentation[36]. Taken as an example, the fragment products as labeled in Fig. 1 can be identified mainly from the following dissociation channels of the doubly charged dimers:

$$Ar_2^{2+} \rightarrow Ar^+ + Ar^+ \tag{1}$$

$$(N_2Ar)^{2+} \rightarrow N_2^+ + Ar^+ \tag{2}$$

$$(N_2)_2^{2+} \rightarrow N_2^+ + N_2^+ \tag{3}$$

However, there is one two-body dissociation channel labeled (4) in Fig. 1 which cannot be simply identified as a direct fragmentation of the vdW dimer $N_2Ar$ according to the above mentioned rules. From calibration we find that one fragment has a charge to mass ratio of 1/14, while the correlated fragment has a charge to mass ratio of 1/54. It is surprising, because none of the dimer components and probable residual gaseous atoms have such a combination of charge to mass ratios. The coincidence

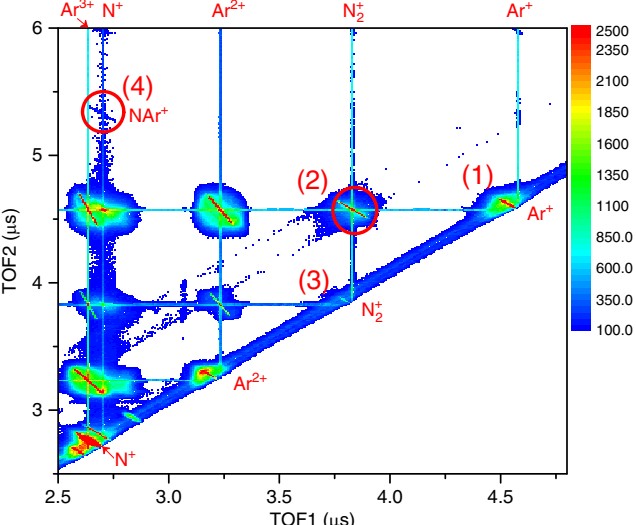

**Fig. 1 Two-dimensional TOF correlation map.** The coincidence map shows the time-of-flights between the first and the second fragment ions arriving at the detector. The product ion species are labeled in the plot for horizontal and vertical axes according to their TOFs, respectively. The corresponding reactions labeled with a number are explained in the text.

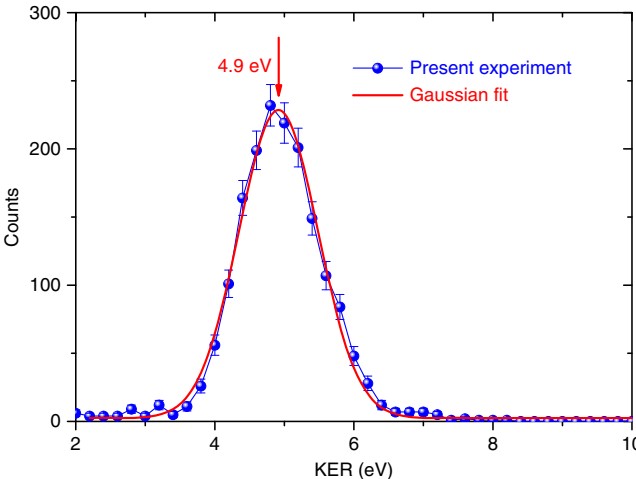

**Fig. 2 Measured KER distributions of N⁺ ion transfer channel.** The blue solid circles are experimental results. The error bars are standard statistical uncertainties. The red curve is a fit to the experimental data, and its peak locates at 4.9 eV. Source data are provided as a Source Data file.

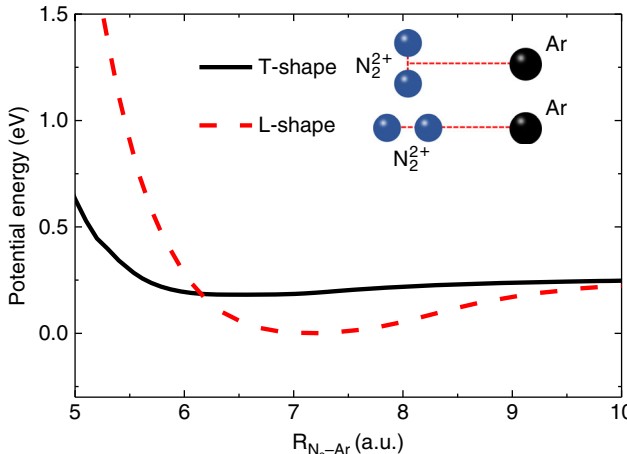

**Fig. 3 Calculated potential energy curves for doubly charged cluster $N_2^{2+}Ar$ ions.** The PECs for T-shaped and linear-shaped $N_2^{2+}Ar$ ions along the $N_2$–Ar bond shown with solid black curve and dashed red curve, respectively. The dimer geometries for T-shaped and linear-shaped structure are schematically displayed.

measurement between $(1/14)^+$ and $(1/54)^+$, charge to mass ratio fragments, suggests that this channel corresponds to the following reaction:

$$(N_2Ar)^{2+} \rightarrow N^+ + NAr^+ \qquad (4)$$

In this reaction channel, a significant discovery is that a heavy $N^+$ ion (mass = 14) is transferred from one molecule center $(N_2^{2+})$ to the other (Ar) in the dimer. One has to note that this exotic dissociation channel means that after the $N_2Ar$ dimer loses two electrons, the strong N–N covalent bond breaks up and, due to the presence of the Ar atom, a new N–Ar bond is formed. We refer to this channel as $N^+$ ion transfer channel. From the potential energy curves (PECs) of $NAr^+$ (ref. [37]), the stable molecular ion $NAr^+$ can only be formed from $N^+$ ion and neutral Ar atom. This suggests that the channel (4) can only originate from the $N_2^{2+}Ar$ parent dimer ion.

**Kinetic energy release of $N^+$ ion transfer channel.** The kinetic energy release (KER) distribution of $N^+$ ion transfer channel shows a single peak at 4.9 eV as illustrated in Fig. 2. In the reaction (4), $N^+$ ion transits from an initially bound $N_2^{2+}$ state to a final $NAr^+$ bound state. On the other hand, the KER results from three contributions: the first one is the polarization energy of $N_2^{2+} - Ar$ bond, which is very weak and can be ignored; the second one is the dissociation energy of $N_2^{2+}$, which is determined from the initial electronic metastable state of $N_2^{2+}$, and the last one is the binding energy of $NAr^+$. According to the PEC of $N_2^{2+}$ molecular ion [38], the $N_2^{2+}$ molecular ion of $N_2Ar$ dimer could be populated in the low-lying electronic states, namely, $X^1\Sigma_g^+$, $a^3\Pi_u$, $D^3\Sigma_u^+$, $A^1\Pi_u$, and $A^3\Sigma_g^-$, etc. However, from the measured KER = 4.9 eV, we infer that the $N^+$ ion transfer channel originates essentially from the relaxation of low vibrational levels of $N_2^{2+}(X^1\Sigma_g^+)Ar$ and $N_2^{2+}(a^3\Pi_u)Ar$ states, because their dissociation energy is less than 4.9 eV. Here, the neutral Ar in the dimer plays a crucial role in the dissociation process of $N_2^{2+}$ $(X^1\Sigma_g^+, a^3\Pi_u)$ dication.

## Discussion
To understand the reaction mechanism of $N^+$ ion transfer channel, let us first consider the PECs of the doubly charged cluster $(N_2Ar)^{2+}$. We mainly focus on the relaxation of

$N_2^{2+}(X^1\Sigma_g^+)Ar$, which should be more important than the relaxation of $N_2^{2+}(a^3\Pi_u)Ar$ due to the fact that the $N_2^{2+}(X^1\Sigma_g^+)$ state is predominantly located at the dissociation energy region allowed by Frank–Condon transition. The PECs for $N_2^{2+}(X^1\Sigma_g^+)Ar$ along the $N_2$–Ar dimer axis are calculated as shown in Fig. 3, see calculation of PECs in Methods. A comparison of the PECs for the linear-shaped and T-shaped $N_2^{2+}(X^1\Sigma_g^+)Ar$ dimer is displayed in Fig. 3. It is clearly shown that the energy minimum for linear-shaped $N_2^{2+}Ar$ locates at $R_{N_2-Ar} = 7.2$ a.u. and is about 0.2 eV lower than the one for T-shaped, (here, $R_{X-Y}$ defines the distance between the Y atom and the center of mass of the X molecule in the whole context). It is well known that neutral $N_2Ar$ cluster has a T-shaped structure[16–18]. When the $Ne^{8+}$ projectile captures two electrons from $N_2$-site of neutral $N_2Ar$ cluster, $N_2^{2+}Ar$ ion should undergo firstly an isomerization process going from an initial T-shape to a linear-shape, and then vibrate in the vicinity of the minimum before dissociation.

With the help of theoretical PECs, we present an intuitive physical picture of the heavy $N^+$ ion transfer reaction. Different from the PEC of the monomer $N_2^{2+}$ (black solid curve in Fig. 4), the PEC of linear-shaped $N_2^{2+}Ar$ along the $N^+$–$N^+$ bond is exhibited by a red dashed curve in Fig. 4, where the influence of neutral Ar atom on the potential energy of $N_2^{2+}$ is considered. It is clearly shown that a new potential well, due to the presence of Ar atom, can be found as the $R_{N-N}$ (upper horizontal axis in Fig. 4) increases, and its minimum locates at $R_{N-N} = 7.8$ a.u., corresponding to the distance $R_{N-Ar} = 7.2$–$7.8/2 = 3.3$ a.u. This clearly suggests that a stable $NAr^+$ molecular ion could be formed.

However, to form the $NAr^+$ ion, the $N^+$ ion, initially bound to $N_2^{2+}$ molecule, has to overcome the barrier $E_c$ (i.e. the breakup of $N_2^{2+}$ covalent bond) and to be trapped in the $NAr^+$ well, which is of course extremely difficult. The only possible pathway is tunneling, namely, the $N^+$ ion from the left potential well of $N_2^{2+}(X^1\Sigma_g^+)Ar$ (in vibration states $v = a$) tunnels into the right potential well of $N^+ + NAr^+$ (the pink dashed arrow in Fig. 4), and the $NAr^+$ is formed in high vibrational levels ($v = b$) (see the green wave line). Then, the Coulomb explosion between $N^+$ and $NAr^+$ ion pair takes place along the blue solid curve as shown in Fig. 4. Due to the strong couplings between Coulomb repulsion and vibration of $NAr^+$ at close distances, the $NAr^+$ at high vibration levels ($v = b$) can be converted to lower vibration states

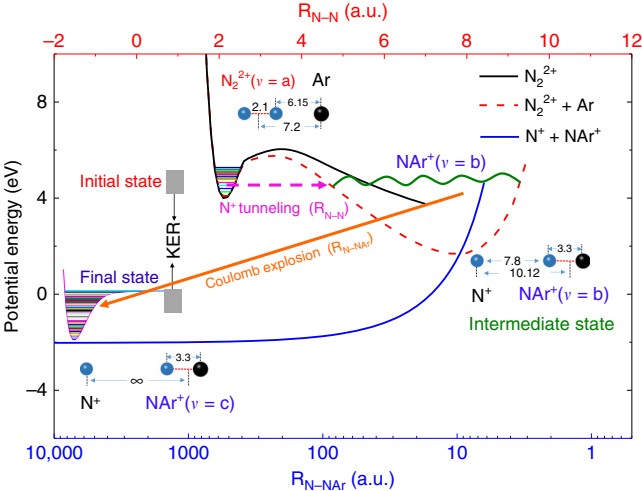

**Fig. 4 Schematic diagram of heavy ion transfer process through tunneling.** The PECs of $N_2^{2+}(X^1\Sigma_g^+)$Ar with/without the presence of neutral Ar along $R_{N-N}$ (upper horizontal axis) and $N^+$–$NAr^+$ along $R_{N-NAr}$ (bottom horizontal axis). The black solid curve for $N_2^{2+}(X^1\Sigma_g^+)$ without the neutral Ar, the red dashed curve for $N_2^{2+}(X^1\Sigma_g^+)$Ar at fixed distance $R_{N_2-Ar} = 7.2$ a.u, and the blue solid curve indicates the PEC of $N^+$–$NAr^+$ with $R_{N-Ar} = 3.3$ a.u. The gray areas indicate the possibly populated initial and final vibrational states of $N_2^{2+}$Ar and $NAr^+$ ions, respectively, which contribute to the heavy ion transfer channel. The final $NAr^+$ potential well is plotted on the left. The pink dashed arrow indicates the $N^+$ ion tunneling. The green wave line indicates the intermediate state of ion $NAr^+$. The origin arrow indicates the fragmentation of $N^+$ and $NAr^+$ ion pair via Coulomb explosion. The origin of the potential energy curves is taken such that the potential energy of $N^+$, $N^+$, and neutral Ar at large separations is zero. The light blue ball represents nitrogen atom while the black ball represents argon atom.

($v = $ c). The correlation between the KER of $N^+$ ion transfer channel and the initial states can be estimated. The initial energy of the doubly charged cluster $N_2^{2+}(X^1\Sigma_g^+)$Ar consists of a sum of minimum potential energy $E_i$ and vibration energy $E_a$ ($v = $ a). Similarly, the final potential energy of the system is given by adding the minimum potential energy $E_f$ and the vibration energy $E_c$ ($v = $ c). On the other hand, as the energy of the system before and after the heavy ion transfer is conserved, as a result, the KER could be obtained as follows: $KER = (E_i + E_a) - (E_f + E_c)$. Namely, $4.9 = (4.0 + E_a) - (-2.05 + E_c)$, that is, $E_c - E_a = 1.15$ eV. Because the final maximum vibration energy is 2.05 eV, while the initial minimum vibration energy is 0 eV, the only possible vibration energies for initial and final states are restricted in the range of 0.0–0.9 and 1.15–2.05 eV, respectively, as shown by the gray areas in Fig. 4.

One has to note that, the monomer $N_2^{2+}$ is a metastable ion which has very long lifetimes. The lifetime of $N_2^{2+}(X^1\Sigma_g^+)$ state for its vibrational level $v = 8$ is as long as 312.5 days ($2.7 \times 10^6$ s) [39], and is even longer for $v < 8$ states. However, for the dimer $N_2^{2+}(X^1\Sigma_g^+)$Ar, the nearby Ar atom modifies the monomer's potential and leads to significant changes of the lifetimes. We calculated the tunneling lifetimes of $N_2^{2+}(X^1\Sigma_g^+)$ in the presence of Ar atom with $R_{N_2-Ar} = 7.2$ a.u. using the complex absorption potential [40], see lifetime estimation in Methods. The calculated tunneling lifetimes of $N_2^{2+}(X^1\Sigma_g^+)$ ($v = 4, 5, 6$) are 1110, 245, and 69 ns (1 ns = $10^{-9}$ s), respectively, which means its lifetime can be reduced by more than 12 orders of magnitude in the doubly charged dimer $N_2^{2+}(X^1\Sigma_g^+)$Ar, from hundred days to nanoseconds. This strongly indicates that, because of the

neighboring Ar atom, the originally quasi-stable monomer ions $N_2^{2+}$ can dissociate rapidly by tunneling through the potential barrier, form the $NAr^+$ ion and, hence, create the $N^+$–$NAr^+$ ion pair.

The heavy ion transfer scenario in a charged dimer may be general for molecular dimer ions such as $AB^{2+}X$ as long as two conditions are met: (i) the metastable covalent bond A–B is toward the neutral particle X, which is the case for most diatomic molecular ions and also for other polar covalent bonds, such as C–H bond; (ii) the neutral particle X and the neighboring ion $A^+/B^+$ can form a bound system $XA^+/XB^+$. This kind of ion transfer process may have potential importance in understanding the underlying mechanisms of biochemistry in tissues, and play an important role in the irradiation damage caused by ion or electron radiation, and furthermore, may directly affect the expression of DNA and protein in vivo.

In conclusion, we present a channel of heavy $N^+$ transfer and $NAr^+$ ion formation in the two-body dissociation of $(N_2Ar)^{2+}$ dimer ion, which results from the breakup of linear-shaped dimer ion $N_2^{2+}(X^1\Sigma_g^+, a^3\Pi_u)$Ar. This transfer mechanism is explained by the tunneling of the heavy $N^+$ ion and, hence, leads to the formation of $NAr^+$ ion. The dissociation of $N_2^{2+}(X^1\Sigma_g^+)$ states through the tunneling of $N^+$ becomes possible due to the presence of Ar atom in the dimer. Consequently, the $N^+$–$N^+$ bond breaks up quickly, and the tunneling of $N^+$ ion and the formation of $N^+$–Ar bond take place almost simultaneously. This is viewed as bound–bound states transition of the heavy ion of $N^+$, followed by Coulomb explosion between $N^+$ and $NAr^+$ ion pairs. Such a mechanism of heavy ion transfer may be general for any molecular dimer ions of type $AB^{2+}X$, and may be of potential importance in various biochemical processes.

## Methods

**Experimental approaches.** Our experiment was conducted at 320 kV platform for multidisciplinary research with highly charged ions in the Institute of Modern Physics, Chinese Academy of Sciences, Lanzhou [41]. A standard Cold Target Recoil Ion Momentum Spectroscopy/Reaction Microscope was employed to determine the momentum of each fragments in coincidence [41–43]. The highly charged $Ne^{8+}$ ions were produced by the permanent magnet electron cyclotron resonance ion source and were accelerated to an energy of 1 MeV. Then the $Ne^{8+}$ ions were delivered to the reaction chamber, where it crossed a supersonic gas target. Various dimers were generated in the molecular jet, such as $N_2$Ar, $(N_2)_2$, and $Ar_2$, by expanding a mixture gas of $N_2$ and Ar through a 30-μm nozzle at 20 bars stagnation pressure under room temperature. The gas pressure ratio in the mixture was $N_2$:Ar ~ 1:1. Charged fragments produced in the collisions were guided by 173 V/cm uniform electric field to a position-sensitive multi-hit detector with a position resolution of 0.1 mm. The information of positions and the TOF of each charged fragment arriving at the detector are obtained; consequently, their three-dimensional initial momentum vectors can be reconstructed by off-line analysis. Downstream, the projectile ions were collected by a Faraday-Cup and the scattered $Ne^{7+}$ and $Ne^{6+}$ ions were detected by a delay-line anode detector. The fragment momentum calibration is carried out in the same conditions by the two-body dissociation of monomer $N_2^{2+}$ ions, which has clearly separated and known peaks [38].

**Calculations of PECs.** The PECs calculations for $N_2^{2+}(X^1\Sigma_g^+)$Ar along the $N_2$–Ar dimer axis are performed using the multi-reference double-excited configuration interaction [44] method with molecular orbitals optimized by complete active space self-consistent field method [45–47]. Here the active space consists of all valence orbitals and totally about 1000 reference configuration state functions are considered in configuration interaction calculations. The orbital wave functions are expanded on aug-cc-pVtZ basis set. The same method is used to calculate the PECs of $N_2^{2+}(X^1\Sigma_g^+)$ and $NAr^+$. Then, the PEC of linear-shaped $N_2^{2+}(X^1\Sigma_g^+)$Ar along the N–N bond is calculated using the two-body potential model [48], where only the two-body correlation interactions are considered based on the PECs of $N_2^{2+}$ and $NAr^+$, and the three-body correlation interaction is neglected. Such a model is reasonable for the PECs calculation of linear-shaped $N_2^{2+}(X^1\Sigma_g^+)$Ar since one of the $N^+$ ions is far away from Ar (>8 a.u.) and the three-body correlation interaction is very weak.

**Lifetime estimation**. Based on the PECs obtained, the lifetimes of $N_2^{2+}(X^1\Sigma_g^+)Ar$ are calculated using the complex absorption potential (CAP) method[49,50]. In the CAP calculations, the nuclear Hamiltonian $H(\eta)$ is extended by a small absorbing potential

$$H(\eta) = -iW(R) + H_0 \tag{5}$$

and

$$W(R) = \begin{cases} 0, & R < R_0 \\ (R - R_0)^2, & R > R_0 \end{cases} \tag{6}$$

in which $\eta$ is a variation parameter and $R_0 = 10$ a.u. is used in the present calculation. A set of $\eta$ values are selected to solve the eigenvalue equation

$$H(\eta)\psi(R) = E(\eta)\psi(R) \tag{7}$$

and the eigenvalue of resonant state can be obtained with the condition of

$$\eta\left|\frac{\partial E(\eta)}{\partial \eta}\right| = \min. \tag{8}$$

Finally, the eigenvalue we obtain is as follows:

$$E = E_0 - 1/2i\Gamma, \tag{9}$$

where $E_0$ is the resonant energy, $\Gamma$ is the resonant width or decay rate, and $\tau = 1/\Gamma$ is the lifetime.

## Data availability

Source data are provided with this paper. The data supporting this study are also available from the corresponding author upon reasonable request.

## Code availability

The code that supports the theoretical plots within this paper is available from the corresponding authors upon reasonable request.

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

## Acknowledgements

This work was supported by the National Key Research and Development Program of China (Grant Nos. 2017YFA0402300, 2017YFA0402400, and 2017YFA0403200), the Science Challenge Project (Grant No. TZ2016005), and the NSFC of China (Grant Nos. 11934004 and 11974358). The authors thank the staff of 320 kV platform for their technical support. The authors acknowledge the helpful discussions with Dr. Bennaceur Najjari and suggestions from Dr. Michael Schulz.

## Author contributions

X.M. and X.L.Z. designed the experiment. X.L.Z., W.T.F., D.L.G., Y.G., S.F.Z., D.M.Z., J.W.X., D.P.D., and B.H. performed the experiment. X.L.Z. and S.C.Y. analyzed the experimental data. X.Q.H., Y.G.P., Y.W., and J.G.W. carried out the theoretical calculations and analysis. X.M., X.L.Z., X.Q.H., A.C., and Y.W. contributed to the interpretation of the data. The manuscript was written by X.M., X.L.Z., X.Q.H., and Y.W. All authors discussed and approved the final version of the manuscript.

## Competing interests

The authors declare no competing interests.
