## [Peer Review File · Nature Communications]

REVIEWERS' COMMENTS

Reviewer #2 (Remarks to the Author):

The authors have addressed my (few) previous concerns. The language has been improved and the introduction has been amended nicely. I would still suggest to slightly change the abstract:

"..a question arises, whether a massive ion could be transferred in biochemical processes and lead to fragmentation? In a complex bio-environment, does heavy ion transfer play a role?" suggests, that the article will address this question. Please add some 'softening' as e.g. "In order to perform a first step towards this topic, small vdw systems can be employed as experimentally feasible model systems."

The compiled listing of previous work on molecular clusters is clearly not complete. This is no problem, but the presentation implies, that there are no other experiments on such systems. Maybe the corresponding sentence can be extended by something like "see for example [...]" or "..rare, some examples can be found in [..]".
..]".

My overall assessment remains unchanged - I support a publication of this nice work in Nature Communications.

Response to the Reviewer #2 (Remarks to the Author)

The authors have addressed my (few) previous concerns. The language has been improved and the introduction has been amended nicely. I would still suggest to slightly change the abstract:

"..a question arises, whether a massive ion could be transferred in biochemical processes and lead to fragmentation? In a complex bio-environment, does heavy ion transfer play a role?" suggests, that the article will address this question. Please add some 'softening' as e.g. "In order to perform a first step towards this topic, small vdw systems can be employed as experimentally feasible model systems."

Reply: Thank you for the good suggestion, we would like to have this in mind. Presently, we have to shorten the abstract and delete the sentences with questions marks according to the formatting requirements of Nature Communications. The updated abstract has 150 words as follows:

Van der Waals clusters are weakly bound atomic/molecular systems and are an important medium for understanding micro-environmental chemical phenomena in bio-systems. The presence of neighboring atoms may open channels although forbidden in isolated atoms/molecules. In hydrogen-bond clusters, proton transfer plays a crucial role, which involves mass and charge migration over large distances within the cluster and results in its fragmentation. Here we report an exotic transfer channel involving heavy N^+ ion observed in a doubly charged cluster produced by 1 MeV Ne^{8+} ions: $(N_2Ar)^{2+} \rightarrow N^+ + NAr^+$. The neighboring Ar atom decreases the N_2^{2+} barrier height and width, resulting in significant shorter lifetimes of metastable molecular ion state $N_2^{2+}(X^1\Sigma_g^+)$. Consequently, the breakup of the covalent N^+-N^+ bond, the tunneling out of the N^+ ion from N_2^{2+} potential well, as well as the formation of $N-Ar^+$ bound system take place almost simultaneously, resulting in a Coulomb explosion of N^+ and NAr^+ ion pairs.

The compiled listing of previous work on molecular clusters is clearly not complete. This is no problem, but the presentation implies, that there are no other experiments on such systems. Maybe the corresponding sentence can be extended by something like "see for example [...]" or "... rare, some examples can be found in [...]".

Reply: Yes, we agree that the suggested presentation is more appropriate. We have changed the citation as "...remains rare, some examples can be found in [16-22]" in the revised manuscript.

My overall assessment remains unchanged - I support a publication of this nice work in Nature Communications.

Reply: Thank you for the encouraging comments!